# Machiavellianism and Gift-Giving in Live Video Streaming: The Mediating Role of Desire for Control and the Moderating Role of Materialism

**DOI:** 10.3390/bs12050157

**Published:** 2022-05-21

**Authors:** Gengfeng Niu, Xiaohan Shi, Siyu Jin, Wencheng Yang, Yang Wu, Xiaojun Sun

**Affiliations:** 1School of Psychology, Central China Normal University, Wuhan 430079, China; niugfpsy@ccnu.edu.cn (G.N.); sxhgloria@mails.ccnu.edu.cn (X.S.); siyoking@mails.ccnu.edu.cn (S.J.); yangwh1@mails.ccnu.edu.cn (W.Y.); 2Key Laboratory of Adolescent Cyberpsychology and Behavior (CCNU), Ministry of Education, Wuhan 430079, China; 3Center for Research on Internet Literacy and Behavior, Central China Normal University, Wuhan 430079, China; 4Collaborative Innovation Center of Assessment toward Basic Education Quality, Central China Normal University Branch, Wuhan 430079, China; 5School of Marxism, Huazhong University of Science and Technology, Wuhan 430074, China

**Keywords:** Machiavellianism, online gift-giving, live video streaming, desire for control, materialism

## Abstract

This study aimed to examine the association between Machiavellianism and gift-giving in live video streaming, as well as the mediating role of desire for control and the moderating role of materialism in this relation. A sample of 212 undergraduate students (146 males; the average age was 19.80 ± 2.05 years old) with experience of gift-giving in live video streaming was recruited to complete questionnaires on Machiavellianism, desire for control, materialism, and the frequency of gift-giving in live video streaming. The results showed that Machiavellianism was positively associated with gift-giving in live video streaming through the mediating role of desire for control; and the mediating effect of desire for control was moderated by materialism, with this relation being stronger for individuals with a higher level of materialism. Though with several limitations (e.g., cross-sectional method), this study could deepen our understanding of the influencing mechanism of gift-giving in live video streaming, which could also provide practical implications for the sustainable development of the live video streaming industry.

## 1. Introduction

With the development of society, especially information technology, the manners of social interaction and entertainment have profoundly changed. In recent years, live streaming platforms (such as Twitch, YouTube Live, and Facebook Live in western countries; AfreecaTV in Korea; YY Live, Douyu TV, and Huya Live in China), where anyone can deliver real-time broadcasts or watch and interact with the streamers [1], have been increasingly popular around the world. The viewing of live video streaming has become one of the most popular online activities [2]; there are 616 million live streaming users in China, representing 62.4 percent of the Chinese Internet population [3]. To some extent, live video streaming has been a new social media, providing users new manners for social interaction and entertainment.

A notable feature of live streaming is that users could interact with the streamer by sending messages and giving paid gifts. Gift-giving is a common feature of most live streaming platforms with specific interpersonal purposes—viewers send gifts to express their encouragement and “like” for the streamer, with the aim of getting special feedback from the streamer (such as streamers’ warm and immediate response), as well as superior social status in the live streaming channel of the streamer [4,5]; in addition, as the gifts are usually purchased with real money from the platforms, gift-giving is also a unique and lucrative business model, from which both streamers and the platforms receive most of their revenues [6,7]. However, too much involvement in giving paid gifts would cause negative social influences on both viewers (e.g., compulsive or irrational tipping could cause economic burden or boast negative beliefs) and the healthy development of the live streaming industry (e.g., negative social reputation and strict industry control). Therefore, it is of vital importance to examine the factors associated with gift-giving in live video streaming, since such findings may have implications for preventing compulsive tipping and fostering the sustainable development of the live video streaming industry.

However, previous studies mainly focused on the factors underlying viewing live video streaming [8]. A number of relevant studies have suggested that the social interaction perspective may offer an integrated view of gift-giving both in real life and online [6,8,9], and Machiavellianism was found to be a significant personality trait directly related to social interaction and motivations in both real life and online space. Based on relevant empirical and theoretical evidence, this study aimed to examine the influencing mechanism of gift-giving in live video streaming. To be specific, it aimed to examine the association between Machiavellianism and gift-giving in live video streaming, as well as the potential mediating role of desire for control and the moderating role of materialism in this relation.

## 2. Literature Review

Though no research examines this issue directly, there are several forms of empirical and theoretical evidence that could provide the basis and foundation for this study and the corresponding hypothesis.

### 2.1. Machiavellianism and Gift-Giving in Live Video Streaming

The attempts to examine the influencing factors of gift-giving in live video streaming are limited. Previous studies mainly focused on the social motivational factors of gift-giving in real life, defining gift-giving as a process that serves communication, social exchange, economic exchange, and socialization functions [7,10]. It has been well established that gift-giving is a common strategy in social interaction. Yang and Urminsky demonstrated that gifting is associated with a desire for recipients’ positive reactions and feedback [9]. Various relevant empirical studies have shown that people often give gifts in order to gain social status or personal attention in the real and the virtual world [11,12]. As in live video streaming, viewers could realize their desire for social interaction through economic input or expenditure on streamers. Therefore, the social interaction perspective may be a new perspective for understanding gift-giving in live video streaming.

Individuals’ social interactions may be greatly influenced by their personal traits, insofar as people with different personal traits may have different motivations and tendencies in daily social interaction and relationships [13]. As one of the three socially aversive or most concerning personality traits, Machiavellianism is characterized by emotional detachment, low empathy, and the tendencies to manipulate or exploit others, which has an important impact on social interaction and relationships in both real life and online space [14,15,16]. Machiavellianism impels individuals to engage in negative behaviors such as bullying, deception, and relational aggression [17], with the aim of increasing social status, limiting the power of others, or dominating the social relationship or interaction [18]. At the same time, they engage in behaviors that benefit themselves at the expense of others, as well as those that give the appearance of success when in fact this is not the case [19].

The process of gift-giving could be seen as a special social interaction between viewers and streamers in live video streaming. When viewers send gifts, the information about the viewer and gifts will be presented in the information bar which all viewers could see, and the streamers will also express their gratitude to the gift-givers in public; afterward, the streamers would also respond warmly to the gift-givers, for example, by answering their questions immediately or conducting specific behaviors (e.g., singing a song or telling a joke) at their request. Thus, it is assumed that this process could satisfy the need to seek social status, and individuals would tend to engage in more gift-giving activities. Relevant studies also revealed that the motivations for social status, social competition, social recognition, and the special attention from streamers were closely associated with gift-giving in live video streaming [5,6,7]. Taking into account the core features of Machiavellianism and relevant findings, it was hypothesized that Machiavellianism would be positively associated with gift-giving in live video streaming (which was assessed through self-reported frequency).

### 2.2. The Mediating Role of Desire for Control

The desire for control refers to the extent to which individuals are motivated to control their environment or exercise dominance over interpersonal situations [20]. The desire for control is a common motivation in daily life, which influences people’s interactions with others. Individuals with a high level of desire for control usually tend to be assertive, decisive, and active. Consequently, they would not only seek to influence others (especially when doing so is advantageous), but also to restrict their interaction partners as well as to use several control-maintaining strategies [21,22]. At the same time, the desire for control was closely associated with the need for social competence [21].

To some extent, live video streaming provides a perfect platform for viewers to satisfy the psychological needs motivated by the desire for control by easily sending gifts. First, to attract more viewers and get more paid-gifts, streamers would adopt various strategies (e.g., giving warm feedback and behaving as required) to make the viewers feel comfortable, influential, and dominant, which could satisfy the psychological needs driven by the desire for control [4,8]. Then, the gift-giving in live video streaming is motivated by several determinants, such as seeking social status and competition, and intentional or unintentional interpersonal manipulation of the streamer (e.g., social recognition, special attention, and appreciation from the streamer) [5,6,7]. Thus, the desire for control would motivate individuals to engage in gift-giving behaviors. Thus, it is hypothesized that the desire for control is positively associated with gift-giving in live video streaming.

In addition, research also indicated that the desire for control was closely associated with Machiavellianism. On the one hand, manipulating or exploiting others in social interaction is a defining element of Machiavellianism [14,15]. Particularly, individuals with a high level of Machiavellianism always attempt to control others by being domineering in multiple social settings [18,19]. On the other hand, individuals with a high level of Machiavellianism tend to control interpersonal interactions and the social environment, aiming to pursue and maintain social power and influence [16,23]. Relevant empirical studies also supported this point and found that Machiavellianism was positively associated with emotional manipulation, supervision, and surveillance (the indicators of desire for control) [14,23,24]. At the same time, the trait activation theory argues that people are attentive to situations and things that activate psychological processes that underlie their personalities [25]. In particular, the trait activation process occurs when the situation or thing is relevant to a person’s values, goals, and the way he or she wants to present himself or herself. As previously discussed, gift-giving in live video streaming fits well into the psychological motivations of Machiavellianism and the desire for control to some extent, and relevant empirical studies also found that subordinate perceptions of authoritarian leadership behavior (a typical manifestation of the desire for control in the workplace) fully mediated the relationship between Machiavellianism and abusive supervision [23]. According to the relevant empirical findings and the main points of the trait activation theory, it was hypothesized that the desire for control would mediate the association between Machiavellianism and gift-giving in live video streaming.

### 2.3. The Moderating Role of Materialism

Materialism is defined as the importance attached to the possessions and the acquisition of material goods in achieving major life goals or desired states [26,27]. It is an important value influencing the way people interpret their environment and construct people’s lives and positions in society [28,29]. Materialism reflects not only the natural attributes of wealth, such as the exchangeability of money, but also the social attributes of wealth, such as social symbolism and compensation of material possessions. For example, individuals with a high level of materialism are more likely to purchase goods excessively and compulsively [30,31]. In particular, individuals with a high level of materialism tend to utilize money or wealth to manage or control social relationships (e.g., compensation for social exclusion and poor/insecure relationships, or pursuit of power) [32,33]. At the same time, materialists may also be prone to pursuing or enhancing social status or power through money or material products [29,34,35].

As discussed above, the nature of paid gift-giving in live video streaming is an exchange for social relationships, attention, power, and status with money, which fits well into the mindset of materialism [32]. Thus, individuals with a high level of materialism may be more likely to engage in gift-giving to satisfy the desire for control induced by Machiavellianism in social interaction. Relevant studies also tested the moderating role of materialism and found that materialism could moderate the relationship between discretionary activities and happiness [36], as well as the effect of accounting for time on prosocial behaviors [37]. On this basis, it was hypothesized that the mediating effect of desire for control in the association between Machiavellianism and gift-giving would be moderated by materialism, and this mediating effect would be stronger among individuals with a high level of materialism.

## 3. The Current Study

To sum up, the present study aimed to examine the association between Machiavellianism and gift-giving in live video streaming, as well as the underlying mechanism—the potential mediating role of desire for control and the moderating role of materialism. In addition, undergraduate students are active in various online activities, and they are also fairly active in live video streaming [3,6,38]. At the same time, the questionnaire was adopted to examine this issue, due to the fact that it was a convenient way to examine the underlying mechanism. Thus, this research aimed to examine this issue among undergraduate students through questionnaires. Based on the above discussion, it was hypothesized that: (1) Machiavellianism would be positively associated with gift-giving in live video streaming; (2) Desire for control could mediate the association between Machiavellianism and gift-giving in live video streaming; (3) Materialism could moderate the mediating effect of desire for control (specifically the pathway between desire for control and gift-giving), and this mediating effect would be stronger among individuals with a high level of materialism.

## 4. Materials and Methods

### 4.1. Participants

Convenience sampling was adopted in this study to recruit undergraduate students (who are also active in live video streaming) to participate in this study. Two hundred and twelve undergraduate students (146 males, and the average age was 19.80 ± 2.05 years old), who have the experience of gift-giving in live video streaming, were recruited from a university in central China to participate in this study. This study was approved by the Ethical Committee for Scientific Research at the researchers’ affiliated institution, and complied with the ethical guidelines protecting human participants. In the study, they were asked to complete a set of scales on the main variables and report relevant information—age, gender, and the frequency of viewing streaming in their daily lives.

### 4.2. Measurement

#### 4.2.1. Machiavellianism

The Mach-IV scale [15], a widely used scale in previous studies, was adopted to measure individuals’ Machiavellianism beliefs with 20 items (e.g., “Never tell anyone the real reason you did something unless it is useful to do so”). These items of this scale focus on cynicism, morality, and manipulative behavior. Participants were asked to indicate the extent to which they agree with the statement of each item on a 7-point scale (1 “strongly disagree”–7 “strongly agree”), with a higher score indicating a higher level of Machiavellianism personality. This scale has been used among Chinese adults with good validity and reliability [39]. In the current study, Cronbach’s alpha for this measure was 0.79.

#### 4.2.2. Desire for Control

The scale developed by Burger and Cooper was adopted to assess individuals’ desire for control [20]. It contains 20 items (e.g., “I enjoy being able to influence the actions of others”), and participants were asked to respond on a 7-point scale (1 “his sentence does not describe me”–7 “this sentence greatly describes me”), and high scores indicate a greater desire for control. It has been translated into Chinese and has good validity and reliability in the current study: the confirmatory factor analysis revealed an acceptable fit: *χ*^2^/*df* = 2.74, RMSEA = 0.05, NFI = 0.95, CFI = 0.97, and Cronbach’s alpha was 0.85.

#### 4.2.3. Materialism

The Chinese version of the Material Value Scale [27] was used in this study, which uses 15 items (e.g., “admire people who own expensive homes, cars, and clothes”) to measure the Materialism value. The participants were asked to respond on a five-point scale (1 “strongly agree”–5 “strongly disagree”), with higher scores indicating a higher level of Materialism. The Cronbach’s alpha for this scale was 0.83 in the current study.

#### 4.2.4. Gift-Giving in Live Video Streaming

An item (“How often do you give paid gifts to the streamers when you are viewing the live video streaming”) was created to measure the frequency of gift-giving in live video streaming, and participants were asked to assess this item on a five-point Likert scale ranging from 1 “rarely” to 5 “always”, and a high score indicates more frequent gift-giving when viewing the live video streaming.

### 4.3. Statistical Analysis

All the statistical analyses were conducted with the SPSS 23.0 (IBM Corp, Armonk, NY, USA). First, the descriptive statistics were computed for the main study variables, and then Pearson’s correlation analysis was also conducted to examine the relationships among the variables. Then, the PROCESS macro [40], which was developed and widely used to test complex models with moderating and mediating effects, was adopted to test the hypothesized moderated mediation model with 5000 bias-corrected bootstrapped samples from the original data. These bootstrapped samples were used to estimate the 95% confidence interval (CI), and the effect is considered significant if the 95% confidence interval values do not include zero. Specifically, Model 4 was conducted to test the mediating model with psychological security as the mediator. Afterward, the Model 14 was conducted to test the integrated model with desire for control as the mediator and materialism as the moderator.

## 5. Results

Table 1 represents the means and standard deviations of the variables, as well as the Pearson’s correlation results among the main research variables—Machiavellianism was positively correlated with desire for control, materialism, and gift-giving in live video streaming; both desire for control and materialism were positively correlated with gift-giving in live video streaming.

Then, the hypothesized moderated mediation model was subsequently tested using Hayes’ the SPSS macro PROCESS with 5000 bootstrapping sampling [40]. Gender, age, and the frequency of viewing live video streaming were all included in the analysis as control variables. Firstly, the simple mediating model analysis showed that, the total effect of Machiavellianism on gift-giving was 0.27 [Boot SE = 0.06; Boot 95% CI = (0.14; 0.40)]; and the indirect of desire for control was 0.15 [Boot SE = 0.06; Boot 95% CI = (0.05; 0.28)], which accounted for 58.87% of the total effect.

The main results of the moderated mediation model analysis consist of two parts: the regression analysis model and the conditional indirect effect analysis, and they are presented, respectively, in Table 2 and Table 3. First, as shown in Table 2, Machiavellianism was positively associated with desire for control; desire for control was positively associated with gift-giving in live video streaming, while Machiavellianism was not significantly associated with gift-giving. These results indicated that desire for control could fully mediate the association between Machiavellianism and gift-giving in live video streaming. Meanwhile, the interaction effect of desire for control and materialism on gift-giving was significant, identifying the moderating role of materialism in the association between desire for control and gift-giving in live video streaming.

Then, as can be seen from Table 3, all the three mediating effects (at the mean of materialism as well as plus and minus one standard deviation from the mean of materialism) were positive and did not include zero. Namely, the indirect effect of Machiavellianism on gift-giving in live video streaming through desire for control was observed among individuals with different levels of materialism. However, the mediating effect of desire for control was stronger for individuals with a higher level of materialism (see Figure 1).

## 6. Discussion

This study examined the psychological mechanisms contributing to gift-giving in live video streaming. First, the results found that Machiavellianism was positively associated with gift-giving in live video streaming, consistent with the findings on Machiavellianism. As one of the “offensive yet non-pathological personalities” [41], Machiavellianism is closely associated with the tendency to manipulate or exploit others in social interaction. Previous studies found that individuals with a high level of Machiavellianism tend to manipulate others and seek social status/power through various strategies, e.g., deception, protective self-monitoring, and self-promotion [14,15]. Despite its importance as a social lubricant [11], gift-giving may also serve egregious purposes, such as manipulation or the corruption of others. According to empirical results, the pursuit of social status, social recognition, and special attention from streamers were heavily associated with gift-giving in live video streaming; and gift-giving may also be considered as a tool to achieve the aims of managing or controlling interaction with the streamers, for example, getting a warm, special, and immediate response [4,5]. In addition, individuals who give paid gifts to the streamer appear to be striving for high social and economic status. Especially those with a high level of Machiavellianism are more likely to develop and maintain relationships with low quality and internalizing and externalizing problems; and at the same time, gift-giving might also be a compensation for their poor relationships in real life [6,7]. Therefore, individuals with a high level of Machiavellianism are not only likely to view video streaming, but the act of gift giving is also positively associated with them.

Furthermore, the results also examined the mechanism underlying this relationship and found that desire for control could mediate the association between Machiavellianism and gift-giving. The desire for control is a common and prominent social motivation driving one to seek control over the environment and events in life [20]. Theoretically, it was associated with Machiavellianism [18,19], because the core feature of Machiavellianism is the tendency to manipulate or exploit others in social situations, and individuals of Machiavellianism are indeed more likely to exert control or dominance over the environment in various settings [19,42]. Furthermore, Machiavellianism was positively associated with the desire for control and controlling behaviors in personal relationships and social interaction [14,23,24]. At the same time, the desire for control also contributes to gift-giving in live video streaming. Individuals with a high level of desire for control are motivated to be assertive, decisive, and active, seeking autonomy and competence, as well as the opportunity to exert influence or control on others [21,22]. Therefore, Machiavellianism might activate individuals’ desire for control and further lead to gift-giving. Though there is a study indicating a negligible association between Machiavellianism and a desire for control, many studies revealed a close association between them with direct or indirect evidence [14,23,24]. Especially in live video streaming, where the social status is unequal, to attract more viewers and get more revenue, the streamer should be agreeable and strive to satisfy viewers’ psychological needs. This finding also fits well with the main points of trait activation theory [25], which argues that people are attentive to situations and things that activate psychological processes underlying their personalities. In this study, gift-giving in live video streaming is a “perfect” situation for individuals with Machiavellianism because it fits well into the psychological motivations and structure of Machiavellianism and desire for control. Thus, Machiavellianism was positively associated with gift-giving in live video streaming through the mediating role of desire for control. Besides, to some extent, this finding also expanded trait activation theory, suggesting it could be applied to the Internet and explain online behaviors. Besides, it should be acknowledged that the association between Machiavellianism and the desire for control among them may be boasted, due to the fact that all the participants watch live video streaming and have the experience of gift-giving, and this relation should be extensively examined in future studies.

In addition, the results also found that materialism could moderate the mediating effect of the desire for control (moderating the association between the desire for control and gift-giving). Especially, the mediating effect of desire for control was stronger for individuals with a higher level of materialism. Materialism refers to the cognitive tendency to how important a person values possessions and acquisition of material goods [26,27]. Individuals with a high level of materialism tend to interpret the outer environment based on material goods, especially money, and firmly hold the belief that “money is power” [28,29]. Accordingly, they would adopt an exchange mindset, believing that all the things in the world could be exchanged for material goods and money. In relevant empirical studies, individuals with a high level of materialism tend to use their wealth or money to manage or control their social relationships (e.g., compensation for social exclusion, poor or unstable relationships, or pursuit of social status and power) [29,34,35]. Namely, they are more likely to satisfy their daily needs (e.g., the desire for control induced by Machiavellianism) through money and material. At the same time, individuals with a high level of materialism also tend to send material gifts to others [43]. Considering the psychological motivations and functions of gift-giving in live video streaming—seeking social competition and social status, as well as managing or controlling the interaction or relationship with the streamer (e.g., special attention and response) [4,5,7]. Materialism could significantly moderate the association between the desire for control and gift-giving, and the mediating effect of the desire for control was stronger for individuals with a higher level of materialism. This finding further reveals the mechanism underlying gift-giving by examining individual differences.

## 7. Implications and Limitations

In the current information era, this study examined the psychological motivations of gift-giving in live video streaming by investigating the association between Machiavellianism and gift-giving in live video streaming and its underlying mechanism. First, this study deepens our understanding of live video streaming by focusing on gift-giving and individuals’ personalities (Machiavellianism). Second, by exploring virtual platforms, these findings extend previous research on gift-giving and provide new insight into the social motivations that underpin it. Thirdly, these results also provide practical implications for the sustainable development of the live video streaming industry. For example, the functions that could satisfy viewers’ psychological needs should be offered by the live video streaming platforms to attract viewers; at the same time, individuals with certain characteristics (e.g., Machiavellianism and materialism) should be paid special attention to, and some measures should also be taken to avoid getting too much involved in paid gift-giving (such as compulsive or irrational tipping). Finally, this study also extended studies on Machiavellianism and trait activation theory, suggesting that they could also influence and explain individuals’ behavior in online space.

Several limitations also should be acknowledged. First, due to the cross-sectional method and the small sample (though the viewers who have the experience of gift-giving are really small), causal inference and good generalizability cannot be achieved [44]. Future work should adopt a longitudinal or experimental research design to strictly confirm the causal nature of the relationship in a larger and more diverse sample. Second, the participants were recruited from Chinese undergraduate students, while live video streaming appeals to individuals of different ages, and gift-giving is saturated with great cultural meanings and cultural differences [45], such that diversified participants from different ages and cultural backgrounds are required in future studies. Third, only one personality trait, Machiavellianism, was adopted in this study, and more personal traits and other underlying mechanisms should be considered in future studies. At the same time, because Machiavellianism consists of different dimensions, future studies may consider the influences of different aspects of Machiavellianism so as to uncover the associations clearly. Finally, the classification of different live streaming types is necessary, as the importance of social interaction and satisfaction of psychological needs were different for different types of live streaming, which could have deepened our understanding of this issue [6]. In addition, because gift-giving was measured by the frequency reported by participants, future studies should adopt other measurements or data resources (e.g., the amount of money spent on gift-giving, or the real data of the live streaming platforms) to make the results more valid.

## Figures and Tables

**Figure 1 behavsci-12-00157-f001:**
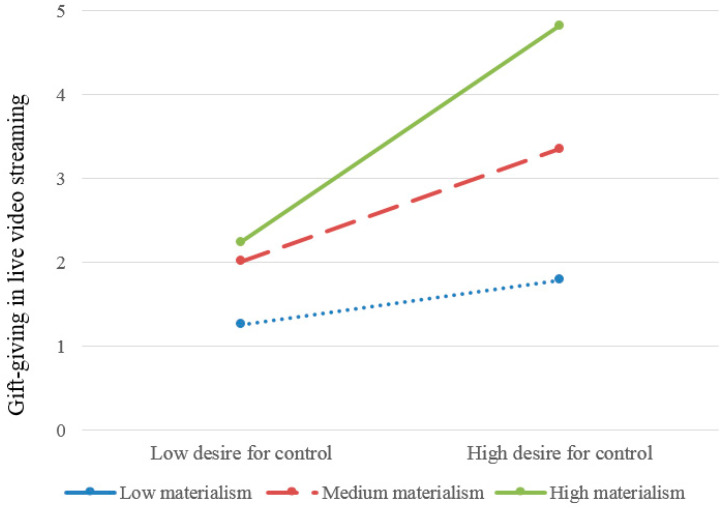
The association between desire for control and gift-giving for different level of materialism.

**Table 1 behavsci-12-00157-t001:** Means, standard deviations, and correlation results.

Variables	*M* (*SD*)	1	2	3	4	5	6	7
1 Gender	-	1						
2 Age	19.8 (2.05)	−0.09	1					
3 Frequency of viewing streaming	3.25 (4.67)	−0.12 *	−0.10	1				
4 Machiavellianism	3.93 (1.28)	0.07	0.09	0.17 *	1			
5 Desire for control	4.27 (1.41)	0.10	0.04	0.15 *	0.65 ***	1		
6 Materialism	3.88 (0.65)	−0.04	0.03	−0.06	0.20 **	0.14 *	1	
7 Gift-giving	2.06 (0.94)	−0.08	0.10	1.12 *	0.32 ***	0.38 ***	0.20 **	1

Note: * *p* < 0.05; ** *p* < 0.01; *** *p* < 0.001; Gender—male “0”, female “1”.

**Table 2 behavsci-12-00157-t002:** The regression analysis of the moderated mediating model.

Dependent Variable	Independent Variable	*R* ^2^	*F*	*β*	Bootstrap LLCI	Bootstrap ULCI	*t*
Desire for control	Gender	0.47	40.09 ***	0.09	−0.01	0.22	0.74
	Age			0.01	−0.04	0.06	0.35
	Viewing frequency			0.03	−0.01	0.05	1.15
	Machiavellianism			0.63	0.55	0.77	12.25 ***
Gift-giving	Gender	0.25	9.74 ***	0.07	−0.23	0.37	0.45
	Age			0.05	−0.02	0.11	1.58
	Viewing frequency			0.09	−0.03	0.17	1.79
	Machiavellianism			0.10	−0.04	0.25	1.85
	Desire for control			0.25	0.12	0.37	4.16 **
	Materialism			0.10	−0.05	0.27	1.82
	Desire for control × Materialism			0.32	0.16	0.49	5.78 ***

Note: ** *p* < 0.01; *** *p* < 0.001; Gender—male “0”, female “1”; LL = low limit, CI = confidence interval, UL = upper limit.

**Table 3 behavsci-12-00157-t003:** The conditional indirect effect analysis.

The Level of Materialism	Mediating Effect	Boot SE	Bootstrap LLCI	Bootstrap ULCI
*M* − 1*SD*	0.09	0.03	0.02	0.19
*M* ± 1*SD*	0.18	0.05	0.04	0.35
*M* + 1*SD*	0.31	0.10	0.08	0.52

Note: LL = low limit, CI = confidence interval, UL = upper limit.

## Data Availability

The data of this study are available from the corresponding author upon reasonable request.

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
