# Peer review of "Machiavellianism and Gift-Giving in Live Video Streaming: The Mediating Role of Desire for Control and the Moderating Role of Materialism"

_behavsci, 2022, doi:10.3390/bs12050157_

Round 1
Reviewer 1 Report
Thank you very much for the opportunity to read the interesting article. I congratulate the authors on a successful text. However, it seems to me that there are several factors that could be improved in the construction of the manuscript. Firstly, I would suggest a more complete presentation of the aim of the article in the context of correlation with the usefulness (scientific and practical) of conclusions that can be drawn from the presented research. In my opinion, there is a clear link between the observations and conclusions and the practical suggestions for the interactive industry. I would also suggest extending the description of the research (research methodology) and including:
- description of the method of selecting the research sample;
- the possibility of extrapolating research results to wider populations, in terms of sample representativeness;
- a more precise description of the research tool used;
- justification of the selection of the research method - in the context of the research objective, the specificity of the research sample and possible empirical limitations.
In conclusion, I would point out more clearly to further possibilities of continuing research, the validity of observing the dynamics of the phenomenon, the existence of trends and comparative possibilities.
Author Response
Thank you very much for the opportunity to read the interesting article. I congratulate the authors on a successful text. However, it seems to me that there are several factors that could be improved in the construction of the manuscript.
Response: Thank you for your kind comments and suggestions, which are of great significance to improving the quality of this manuscript.
Firstly, I would suggest a more complete presentation of the aim of the article in the context of correlation with the usefulness (scientific and practical) of conclusions that can be drawn from the presented research. In my opinion, there is a clear link between the observations and conclusions and the practical suggestions for the interactive industry.
Response: According to your suggestion, we clearly presented the aim and the implications of this study in the Abstract, Introduction, and Discussion.
I would also suggest extending the description of the research (research methodology) and including:
- description of the method of selecting the research sample;
Response: The convenience sampling was adopted in this study. Especially, the students who attended a public elective course were recruited voluntarily to participate in this study, with the requirement that they have the experience of gift-giving in live video streaming; students who agree to participate in this study were asked to complete an online questionnaire. Through this procedure, two hundred and twelve undergraduate students (146 males and the average age was 19.80 ± 2.05 years old), who have the experience of gift-giving in live video streaming, were recruited from a university in central China to participate in this study.
- the possibility of extrapolating research results to wider populations, in terms of sample representativeness;
Response: Previous studies suggest that undergraduate students are active in various online activities, and they are also fairly active in live video streaming. Based on these, this study was conducted among undergraduate students. However, diversified participants from different ages and cultural backgrounds are required in future studies, which could validate and promote the extensibility of the results. This was addressed in the Discussion.
- a more precise description of the research tool used;
Response: According to your suggestion, we extended the contents of this section, with the aim to describe the research tool more precisely.
- justification of the selection of the research method - in the context of the research objective, the specificity of the research sample, and possible empirical limitations.
Response: As you suggested, we provided relevant information to justify the selection of the research method. At the same time, the limitations were also addressed in the Discussion.
In conclusion, I would point out more clearly to further possibilities of continuing research, the validity of observing the dynamics of the phenomenon, the existence of trends and comparative possibilities.
Response: As you suggested, based on this study and relevant evidence, several future directions were clearly pointed out. For example, longitudinal or experimental research design, diversified participants from different ages and cultural backgrounds, and other data resources should be adopted, and more influencing factors and underlying mechanisms should be considered in future studies, all of which could validate and deepen current study.
Reviewer 2 Report
The paper corresponds to the Journal areas. The theme is interesting and actual in the ongoing digitalization of all economic sectors. However, before publishing the paper should be corrected:
Abstract:
- it should be better to avoid symbols Mage = 19.80, SDage = 2.05
- it should be better to clarify the limitations and implications of the study.
Introduction:
- add the paper structure
- clarify why it is actually to analyse under Machiavellianism.
- It would be better to present separately introduction and Literature review
Materials and Methods:
- please, clarify the socio-demographic portrait of the respondents.
- Besides, it would be better to add a list of questions and descriptive statistics of the generated data.
- Furthermore, the paper would benefit if the authors add the justification of chosen methods for analysis (the regression analysis of the moderated mediating model, the conditional indirect effect analysis and etc)
The authors should add information on Patents or delete it.
Besides, the authors should extend the literature review and pay attention to citations and references. As not all references were cited in the paper.
The English should be polished.
Author Response
Comments and Suggestions for Authors
The paper corresponds to the Journal areas. The theme is interesting and actual in the ongoing digitalization of all economic sectors. However, before publishing the paper should be corrected:
Response: Thank you for your kind comments and suggestions, which are of great significance to improving the quality of this manuscript.
Abstract:
it should be better to avoid symbols Mage = 19.80, SDage = 2.05
Response: As you suggested, these symbols were deleted.
it should be better to clarify the limitations and implications of the study. (
Response: According to your suggestion, the limitations and implications were clarified in the Abstract.
Introduction:
add the paper structure
Response: According to your suggestion, we used an independent part “the current study” to describe the main points of the current study.
clarify why it is actually to analyse under Machiavellianism.
Response: Based on relevant evidence, this study aimed to examine the influencing factors and inner mechanism of gift-giving in live video streaming, under the perspective of social interaction. Based on this research perspective and evidence, the factors and hypothesized model were constructed. Machiavellianism has a direct and important impact on social interaction and relationships, and the social motivations in both real life and online space. Thus, this study examined this issue from the perspective of Machiavellianism. According to your suggestion, this idea was highlighted in the revised manuscript.
It would be better to present separately introduction and Literature review
Response: The Introduction and Literature review were separately presented, with the aim to clearly present the research question and relevant evidence.
Materials and Methods:
please, clarify the socio-demographic portrait of the respondents.
Response: As you suggested, the socio-demographic portrait of the respondents collected in this study was reported. All the participants were undergraduate students with 146 males, and the average age was 19.80±2.05 years old.
Besides, it would be better to add a list of questions and descriptive statistics of the generated data.
Response: Thank you for your suggestion! The main aim of this study was to examine the association between Machiavellianism and gift-giving in live video streaming, as well as the underlying mechanism. Thus, we presented the information of the questionnaire items in the Measurement, and the descriptive statistics of each main variable were presented in Table 1.
Furthermore, the paper would benefit if the authors add the justification of chosen methods for analysis (the regression analysis of the moderated mediating model, the conditional indirect effect analysis and etc)
Response: According to your suggestion, we added Statistical Analysis to describe the justification of the analysis and the analysis procedure.
The authors should add information on Patents or delete it.
Response: We deleted it in the revised manuscript.
Besides, the authors should extend the literature review and pay attention to citations and references. As not all references were cited in the paper.
Response: Thank you for your careful review. There is a mistake that some references were presented in the wrong place. According to your suggestion, we checked and revised the references.
The English should be polished.
Response: As you suggested, we read through the manuscript critically to improve the language of this manuscript. At the same time, we also asked a colleague, who worked in English-speaking countries, to help us to polish the language, with the aim to improve the readability and fluency of this manuscript.
Round 2
Reviewer 2 Report
The authors considered the remarks.